# Dietary Intake and the Neighbourhood Environment in the BC Generations Project

**DOI:** 10.3390/nu14224882

**Published:** 2022-11-18

**Authors:** Rachel A. Murphy, Gabriela Kuczynski, Parveen Bhatti, Trevor J. B. Dummer

**Affiliations:** 1School of Population and Public Health, University of British Columbia, Vancouver, BC V6T 1Z3, Canada; 2Cancer Control Research, BC Cancer, Vancouver, BC V5Z 1L3, Canada; 3Faculty of Health Sciences, Simon Fraser University, Burnaby, BC V5A 1S6, Canada

**Keywords:** dietary intake, diet quality, built environment, geography

## Abstract

Poor diet is a major risk factor for many chronic diseases including cancer. Understanding broader contextual factors that influence dietary intake is important for making tangible progress towards improving diet at the population level. This study investigated neighbourhood social and built environment factors and fruit and vegetable intake among ~28,000 adults aged 35–69 years within the BC Generations Project. Daily fruit and vegetable intake was categorized according to guidelines (≥5 servings/day vs. <5 servings/day). Geospatial characteristics included walkability, greenness, marginalization, and material and social deprivation, reflecting access to goods and amenities and social relationships. Generalized, linear mixed-effect models adjusted for sociodemographic factors and lifestyle variables were used to estimate the odds ratios (ORs). Participants living in neighbourhoods with greater material deprivation (e.g., OR = 0.77; 95% CI: 0.70–0.86 for very high material deprivation) and very high social deprivation (OR = 0.90; 95% CI: 0.82–0.99) were less likely to meet recommendations for fruit and vegetable consumption relative to those living in areas with very low material deprivation and very low social deprivation, respectively. Relative to participants living in areas with very low greenness, participants living in neighbourhoods with high (OR = 1.10, 95% CI 1.01–1.20) to very high (OR = 1.11, 95% CI 1.01–1.21) greenness were more likely to meet recommendations for fruits and vegetables. These findings highlight the complexity of dietary intake which may be shaped by multiple neighbourhood characteristics.

## 1. Introduction

Poor diet is a leading risk factor for chronic diseases, death and disability globally [1]. In Canada, 233,900 new cancer cases are projected to be diagnosed in 2023 [2] and 42% of Canadians will be diagnosed with cancer during their lifetime [3]. Estimates from 2015 suggest that approximately 4.7% of cancer cases in Canada are attributable to low vegetable consumption while 7.3% are attributable to low fruit consumption, further underscoring the importance of a healthy diet. However, most Canadians consume low quality diets that are inconsistent with recommendations from Canada’s Food Guide that emphasize eating a variety of vegetables and fruits, choosing whole grains and protein foods while limiting foods high in sugar, sodium or saturated fat [4]. National data from the Canadian Community Health Survey (CCHS)—Nutrition has consistently shown poor alignment with dietary recommendations [5] with little improvement over time [5].

The CFG-2019 and Health Canada’s refreshed Healthy Eating Strategy [4] largely focus on supporting individuals to make informed, healthy choices. While individual behaviours influence dietary intake, it has been posited that the global obesity epidemic is driven mainly by shifts in global food systems and the local environment [6], meaning that diet quality is influenced by factors beyond an individual’s control, including income and social status, employment, education, childhood experiences, the physical environment and other factors collectively referred to as social determinants of health [7]. Evidence from our group and others has shown that differences in social position are reflected in diet quality [8,9]. Interventions to improve dietary intake, therefore, need to consider broader contextual factors, although the best approaches to achieve this while concurrently addressing health inequalities is currently unclear.

The socio ecological model [10] and the framework put forth in the Commission on Social Determinants of Health (CSDH) [11], conceptualize health as being affected by the individual, group/community, physical, social and political environments, as well as interactions between these factors. The neighbourhood social and built environments are intermediary determinants of health, bidirectionally shaped by the structural (policies) and social determinants of health inequities (e.g., income, education and occupation). These intermediary determinants of health are one of the primary mechanisms through which an individual’s exposure and vulnerability to conditions (in this case, diet) are influenced.

Evidence on characteristics of the built environment suggest that people who live in urban environments characterized by greater sprawl, greater density, lower walkability and greenness are more likely to be obese [12,13,14]. However, few studies have considered the potential influence of the built environment on dietary intake beyond measures of the food environment [15]. Studies of the impact of food environment on diet quality have predominately focused on food access; using measures such as proximity to the nearest food store or direct observations of pricing/availability in stores have found inconsistent results [16,17]. Given the complexity of contextual drivers of diet quality, a much more comprehensive consideration of broader determinants of health inequities in the built environment is warranted to drive necessary shifts in diet to improve the health of Canadians.

The social environment encompasses the groups to which people belong, the neighbourhoods where people live, the organization of workplaces and policies people create in their lives [18]. The definition and measurement of social environment constructs are highly variable [18,19], at times cutting across aspects of the built environment and structural determinants of health inequities. The social environment encompasses proximal and distal constructs of the combined roles of society and economy [18]. A limited number of studies suggest that social support at the neighborhood level may also shape diet quality, whereas greater support and resources may be associated with healthier diets [20,21].

Most evidence on dietary intake and the social and built environment is drawn from countries outside of Canada, which may, therefore, not be generalizable to Canada due to sociocultural, climatic, and geographic differences. Studies in Canada have typically focused on single urban areas with limited measures of the environment [22,23,24]. Given the complexity of contextual drivers of dietary intake, recognition and understanding of links between broader measures of the built environment and diet are critically needed. This study aimed to improve our understanding of how built and social environments—specifically walkability, greenness, socio-economic marginalization, and material and social deprivation—influence diet among Canadians, using data from a large population-based cohort.

## 2. Materials and Methods

### 2.1. Study Sample

Participants were drawn from the BC Generations Project (BGCP), a study of 29,850 men and women in British Columbia, Canada [25]. Participants were enrolled between 2009 and 2015 and were 35 to 69 years of age at time of enrollment. All participants provided written informed consent. Ethics approval for this analysis was granted by the University of British Columbia Clinical Research Ethics Board (#H22-00880).

Of the 29,246 participants that completed the baseline health and lifestyle questionnaire, those missing postal code information (*N* = 251) or data on fruit and vegetable consumption (*N* = 698) were excluded. Participants reporting implausible levels of fruit and vegetable consumption were also excluded (i.e., ≥30 servings per day, *N* = 28), leaving a total of 28,269 participants. Sample sizes varied between individual analysis depending on the exposure of interest; 27,667 for walkability and the Canadian marginalization index (CAN-Marg), 28,156 for residential greenness, 28,066 for access to employment and 28,074 for social and material deprivation which were assessed as described below.

### 2.2. Dietary Intake

The baseline questionnaire captured self-reported data on the typical daily number of servings of fruits and vegetables consumed by participants. The frequency of fruit and vegetable consumption reflects diet quality and can be used when data necessary to construct the healthy eating index is not available [26]. The number of servings of fruits and vegetables per day was examined as the outcome variable as a categorical variable based on recommendations to consume ≥5 servings of fruits and vegetables per day for overall health [27] and cancer prevention [28].

### 2.3. Built and Social Neighbourhood Environment

Participants in the BCGP were linked, at the postal code level, to geospatial data collated and generated by the Canadian Urban Environment Health Research Consortium (CANUE). Environmental exposure datasets were linked to CanPath baseline data using participants’ 6-digit residential postal code at time of enrollment. ArcGIS was used to link postal codes to Statistics Canada dissemination areas (DA). DA is the smallest standard geographic area for which census data in Canada are disseminated, containing between 400 to 700 persons. Measures of the neighbourhood built and social environment were available for select years. For this analysis, geospatial data from the year closest to time of baseline questionnaire completion were assigned to participants.

We focused on those factors that were least suggestively associated with dietary intake in prior studies [9,13,14,15,29], including walkability, residential greenness, CAN-Marg, social and material deprivation. Walkability was generated using the Canadian Active Living Environments (Can-ALE) index, which is the sum of z-scores for intersection density and dwelling density that had been measured at the DA level. This variable was subsequently categorized as an index on a scale of 1 (very low walkability) to 5 (very high walkability) [30]. Residential greenness was assessed using the annual mean normalized difference vegetation index (NDVI). NDVI is a measure of vegetation health and is calculated as the difference between the near infrared wavelength reflectance (NIR) and the red wavelength reflectance (RED), divided by the sum of NIR and RED. NDVI measures range from −1 to 1, which we categorized as quintiles based on the distribution of the variable in the study population. CAN-Marg, which was also categorized as quintiles, is a measure combining indicators from four distinct dimensions of marginalization assessed at the DA level: (1) households and dwellings (types and density of housing and family structure e.g., percent living alone, percent of dwellings not owned), (2) material resources (access to and attainment of material needs, e.g., percent unemployed, percent with education below high school degree), (3) age and labour force, e.g., percent of people ≥65, dependency ratio (≥65 years: 15–64 years), percent not participating in the labour force, and (4) immigration and visible minority: percent of recent immigrants, percent who self-identify as a visible minority [31]. Social and material deprivation were measured using six indicators from the Census as described in Pampalon et al. [32]. Principal component analysis subsequently reduced the variables into two components—social deprivation (proportion of single-parent families, proportion of people living alone and proportion of people who are separated, divorced or widowed) and material deprivation (the proportion of people with less than a high school diploma, mean individual income, employment rate). The material and social deprivation scores are categorized as very low (1) to very high (5) within the respective region. 

### 2.4. Covariates

Data on age, sex, ethnicity, marital status, smoking status, alcohol consumption, physical activity, individual level education and household income were drawn from the BCGP baseline questionnaire. Body mass index (BMI) was calculated from in-person assessment measures of height and weight or self-reported height and weight if measured data were not available. A ‘missing’ category was created for participants with missing information for a given covariate. Age was categorized as 35–44 years, 45–54 years, 55–64 years and ≥65 years. Ethnicity was dichotomized as white and non-white due to the small number of participants from individual ethnic groups in BCGP [25]. Marital status was dichotomized as living with or without a partner. Smoking was categorized as never, former or current. Alcohol consumption was categorized as never or former (no alcohol over past 12 months), occasional (≤2–3 times/week) or habitual (≥4–5 times/week). Physical activity was assessed with the International Physical Activity Questionnaire (IPAQ) [33], which asked about the number of days and time spent doing vigorous and moderate activity, walking and sitting in the previous seven days. Physical activity was then categorized as low, moderate and high using the IPAQ scoring protocol [34]. Individual level indicators of socioeconomic status (education and household income): Education was categorized as high school or less, college, and university or higher. Household income was categorized as <$25,000, $25,000 to $49,999, $50,000 to $74,999, $75,000 to $150,000 and >$150,000. BMI was categorized as <25 kg/m^2^, 25–29.9 kg/m^2^ or ≥30 kg/m^2^.

### 2.5. Statistical Analysis

Mixed-effect models were constructed to account for the non-independence of observations or the nested data structure since participants are nested within forward sortation areas (FSAs). FSAs were selected as the level of clusters in the models to achieve sufficient sample size. Generalized, linear mixed-effect models were used to estimate the odds ratios (ORs) of meeting dietary recommendations for fruits and vegetables. A random intercept was included in models to account for area-level variation in dietary intake. Linearity in models was confirmed via residual plots that showed random dispersion [35].

Model 1 was adjusted for sociodemographic factors; sex, age, ethnicity, and marital status. Model 2 additionally adjusted for lifestyle variables; smoking, alcohol consumption, physical activity and BMI. Individual-level income and education and urbanicity (urban areas with census metropolitan areas of at least 100,000 of which 50,000 or more live in the core versus urban areas or census agglomerations [36]) were explored as potential effect modifiers using cross-product terms in models. Statistical analyses were performed with STATA, version 14.2 (StataCorp, College Station, TX, USA) with significance set at *p* < 0.05.

## 3. Results

The demographic, socioeconomic, lifestyle, and residential characteristics of the BCGP cohort sample with complete data on fruit and vegetable consumption and residential greenness are shown in Table 1. Participants were predominately female, of white ethnicity and were likely to live with a partner. The majority of participants had a household income ≥$50,000 and education at or above college level. More than half of participants were never smokers, whereas the majority reported at least occasional alcohol consumption. Females were more likely to consume the recommended servings of fruits and vegetables per day. Upwards of 40% of participants who consumed <5 servings of fruits and vegetables per day reported high levels of physical activity while 53% of participants who consumed ≥5 servings of fruits and vegetables per day reported high physical activity. Being overweight and obesity was prevalent in both groups, most notably among those reporting <5 servings of fruits and vegetables per day (51.6%).

The mean (SD) servings of fruits and vegetables per day among men and women were 4.48 (2.48) and 5.61 (2.59). Overall, 42.5% of men and 34.8% of women consumed at least five servings of fruits and vegetables per day. The prevalence of participants meeting fruit and vegetable recommendations and the mean daily intake of fruit and vegetables by categories of neighbourhood environment are shown in Appendix A.

The odds ratios (OR) of meeting recommendations for fruit and vegetable intake by characteristics of the neighbourhood environment are shown in Table 2. There were no associations between walkability and meeting the recommendation for fruit and vegetable consumption or with dimensions of marginalization aside from age and labour force. Participants living in neighbourhoods with moderate to very high indices of age and labour force, reflecting a greater proportion of seniors and less participation in the labour force, were less likely to meet the recommendation for fruit and vegetable consumption. For example, OR = 0.83 (95% CI 0.73–0.94) for very high relative to very low. Associations were attenuated with additional adjustment for lifestyle variables in Model 2 but remained statistically significant. Participants living in neighbourhoods with greater material deprivation and (moderate to very high) and very high social deprivation were less likely to meet recommendations for fruits and vegetables even with full adjustment for lifestyle variables. For example, relative to very low material deprivation, the OR (95% CI) for very high material deprivation was 0.77 (95% CI 0.70–0.86) in Model 2. Relative to very low greenness, participants living in neighbourhoods with high to very high greenness had greater odds of meeting recommendations for fruits and vegetables, which persisted with adjustment for lifestyle variables (Model 2; high OR = 1.10 (95% CI 1.01–1.20), very high OR = 1.11 (95% CI 1.01–1.21).

Household income and participant level education did not modify associations between fruit and vegetable consumption and characteristics of the neighbourhood social and built environment (*p*-interaction terms > 0.05). Associations between fruit and vegetable consumption, the households and dwellings dimension of marginalization, and material deprivation were modified by urbanicity (*p*-interaction < 0.05). In rural areas, low, moderate and high residential instability (households and dwellings) were associated with lower odds of meeting recommendations for fruit and vegetable consumption (Table 3). No associations were observed in urban areas. Associations between material deprivation and fruit and vegetable consumption were apparent in urban but not rural areas (Table 3). For example, relative to neighbourhoods with very low material deprivation, participants living in neighbourhoods with high and very high material deprivation were less likely to meet recommendations for fruit and vegetable consumption (OR = 0.85, 95% CI 0.77–0.95 and OR = 0.72, 95% CI 0.63–0.82).

## 4. Discussion

Using data from a large population-based cohort study in British Columbia, we observed that participants who lived in neighbourhoods characterized by high to very high greenness were more likely to meet recommendations for fruit and vegetable consumption. Conversely, participants who lived in neighbourhoods with very high social deprivation, moderate to very high material deprivation and moderate to very high indices of age and labour force were less likely to meet recommendations for fruit and vegetable consumption. The relationship between fruit and vegetable intake and material deprivation was most apparent in urban areas, while residential instability was associated with fruit and vegetable consumption only in rural areas. Given that nearly all Canadians would benefit from an increase in consumption of fruits and vegetables [37], and the impact of poor diet on health [1], our findings highlight the need for continued public health action to improve dietary intake.

The finding that greater neighbourhood material deprivation, as assessed by the Pampalon index [32], was associated with lower fruit and vegetable intake aligns with previous findings from our group as well as others [38]. However, a systematic review of health behaviours and neighbourhood deprivation reported inconsistent associations with fruit and vegetable consumption and highlighted the need for more research [39]. It is notable that there were no associations between fruit and vegetable consumption and the material resources dimension of the CAN-Marg measure in our study. Although similarly named, these indices differ in the variables used to construct them and the context of the score. Pampalon’s index considers the proportion of people without a high school diploma, the average individual income and the employment rate [32]. The CAN-Marg measure also includes the proportion of people without a high school diploma, as well as proportions of lone-parent families, government transfer payments, unemployment 15+, below low income cut-off and homes needing major repair [31]. It has been posited that heterogeneous definitions of deprivation, which invariably include material and social deprivation and, as in this study, different measures of each, may contribute to unclear links between fruit and vegetable consumption as well as health behaviours more broadly, and neighbourhood deprivation [9,39]. Operationalization of standard definitions of neighbourhood deprivation is challenging given the need to rely on available data. In the absence of such an approach, careful consideration of the selection of indicators to measure deprivation is needed to facilitate public health monitoring of inequalities.

There are limited previous studies of greenness and dietary intake. A study in New Zealand reported no overall association between greenspace and fruit and vegetable consumption, although fruit and vegetable consumption was lower in urban versus rural areas that had a greater percentage of green space [40]. In our study, meeting fruit and vegetable consumption recommendations was associated with higher greenness. Drawing on the socio-ecological framework, neighbourhood green space represents a ‘settings’ level factor that may intersect and converge to influence eating behaviour [41]. Greenness can be modified via addition of parks, or removal for urbanization or development of commercial and residential areas [42], suggesting potential approaches for intervention. However, it is important to note that greenness in our study was defined using NDVI and not access to greenspace. The use of NDVI to measure greenness may result in inflated values in agricultural areas [43], although the majority of our population resided in urban areas and urbanicity did not modify associations between fruit and vegetable consumption and NDVI.

In this study, moderate, high and very high scores on the age and labour force dimension of marginalization were associated with lower intake of fruits and vegetables. The age and labour force dimension provides a measure of neighbourhood-level dependency; the proportion of seniors, the dependency ratio (seniors and children to population 15–64) and proportion not participating in the labour force. The inverse association between fruit and vegetable intake and dependency aligns with findings of dependency and health and social outcomes. A study in Ontario reported inverse gradients between quintiles of dependency and overall mortality, mental health emergency department visits and alcohol retail locations [44]. We are unaware of prior studies that have examined this dimension of marginalization with dietary intake or health behaviours, and it is unclear whether this is a determinant, correlate or intervening variable with respect to dietary intake. Sawyer et al. [45] have proposed that dietary intake reflects a much more complex adaptive system than previously posited. Applying their proposed systems map, it is possible dependency shapes the objective food environment, including supply, food costs and geographical accessibility, or reflects a subsystem that influences food intake such as social and cultural influences (i.e., social relationships, social model of consumption). In future work, exploration of food environment characteristics, in addition to the social and built environment, may help to clarify relationships.

A strength of this study was the availability of numerous indicators of the social and built environment. This enabled novel exploration of associations with fruit and vegetable intake. The methodology used to measure the neighbourhood environment indicators was also rigorous, drawing on widely used measures and best practices [46,47]. A further strength is the large population of men and women from diverse neighbourhoods across British Columbia, including all seven census metropolitan areas (Vancouver, Victoria, Kelowna, Abbotsford, Nanaimo, Kamloops and Chilliwack). The sample population also included rural neighbourhoods, which facilitated exploration of urbanicity as an effect modifier. There are, however, several limitations of our study. Dietary intake was limited to fruit and vegetable consumption which, although correlated with dietary quality [26], provides limited insight on overall dietary intake which may be more relevant for health and chronic disease prevention. Fruit and vegetable consumption was self-reported which may be prone to bias. Further, consumption was limited to the typical amount consumed at present which may not reflect earlier consumption patterns that are relevant for disease prevention. The cross-sectional design precludes the ability to determine causality. The neighbourhood environment was based on where people lived, which may differ from where people work, play or study. It is thus possible the characterization of neighbourhood environment and fruit and vegetable consumption did not encompass all environments that shape behaviours for some participants. Lastly, the timing of the assessment of dietary intake and neighbourhood characteristics did not always coincide, as some measures of the environment were drawn from Census data, which is only collected in Canada every four years. There is thus the potential for misclassification; however, characteristics of neighbourhoods such as marginalization are generally stable overtime [44].

## 5. Conclusions

In this population, dietary intake, specifically fruit and vegetable consumption at the individual level, was associated with characteristics of the neighbourhood built and social environments. The findings underscore the complexity of dietary intake, and the importance of considering potential influential factors of diet beyond the immediate food environment. When possible, characterizing socio environmental factors such as those identified herein, greenness, social and material deprivation, when implementing interventions aimed at improving dietary intake may help facilitate success. The findings also suggest potential targets for policies and interventions to improve dietary intake as inequalities were particularly evident in neighbourhoods with lower access to and attainment of basic material needs. Additional studies with detailed dietary data, and more diverse populations would help to further understand the role of the neighbourhood environment on dietary intake.

## Figures and Tables

**Table 1 nutrients-14-04882-t001:** Characteristics of the BCGP cohort sample ^1^.

Characteristics	<5 Servings Fruit & Vegetables/Day *N* = 11,951 (41.9%)	≥5 Servings Fruit & Vegetables/Day *N* = 16,597 (58.1%)
**Age, *N* (%)**		
35–44 years	1611 (13.6)	2158 (13.1)
45–54 years	3270 (27.6)	4386 (26.7)
55–64 years	4677 (39.5)	6778 (41.3)
≥65 years	2275 (19.2)	3108 (18.9)
**Sex, *N* (%)**		
Male	5044 (42.8)	3724 (22.8)
Female	6741 (57.2)	12,647 (77.3)
**Marital status, *N* (%)**		
Living with a partner	8775 (74.5)	12,500 (76.4)
Living without a partner	2954 (25.1)	3798 (23.2)
Missing	56 (0.48)	73 (0.45)
**Ethnicity, *N* (%)**		
White	9937 (86.9)	14,627 (91.9)
Non-white	1496 (13.1)	1285 (8.08)
**Income, *N* (%)**		
$0–24,999	802 (6.81)	725 (4.43)
$25,000–49,999	1960 (16.6)	2442 (14.9)
$50,000–74,999	2474 (21.0)	3209 (19.6)
$75,000–149,999	4242 (36.0)	6401 (39.1)
>$150,000	1626 (13.8)	2612 (16.0)
Missing	681 (5.78)	982 (6.00)
**Education, *N* (%)**		
≤High school	2688 (22.8)	2719 (16.6)
College	4747 (40.3)	5934 (36.3)
≥Bachelor’s degree	4280 (36.3)	7640 (46.7)
Missing	70 (0.59)	78 (0.48)
**Smoking status, *N* (%)**		
Never	5890 (50.9)	9049 (56.0)
Former	4837 (41.8)	6663 (41.2)
Current	720 (6.22)	323 (2.00)
Missing	126 (1.09)	133 (0.82)
**Alcohol consumption, *N* (%)**		
Never/Former	1244 (10.6)	1543 (9.43)
Occasional	7120 (60.4)	10,289 (62.9)
Regular	3367 (28.6)	4495 (27.5)
Missing	54 (0.46)	44 (0.27)
**Physical activity, *N* (%)**		
Low	2339 (19.9)	1774 (10.8)
Moderate	3959 (33.6)	4970 (30.4)
High	4796 (40.7)	8699 (53.1)
Missing	691 (5.86)	928 (5.67)
**BMI,** **Mean (SD)**		
<25.0 kg/m^2^	3987 (33.8)	7152 (43.7)
25.0–29.9 kg/m^2^	3828 (32.5)	4854 (29.7)
≥30.0 kg/m^2^	2246 (19.1)	2531 (15.5)
Missing	1724 (14.6)	1834 (11.2)
**Urbanicity, *N* (%)**		
Urban	8417 (71.6)	11,658 (71.4)
Rural	3333 (28.4)	4661 (28.6)

^1^ Characteristics are shown for the *N* = 28,156 with data on residential greenness.

**Table 2 nutrients-14-04882-t002:** Association of built and social environment characteristics with fruit and vegetable consumption in the BC Generations Project.

	OR (95% CI)	
	Model 1	Model 2
**Walkability (Can-ALE)**		
Very low	Ref.	Ref.
Low	1.05 (0.98–1.12)	1.05 (0.98–1.13)
Moderate	0.98 (0.90–1.06)	0.99 (0.91–1.08)
High	1.06 (0.96–1.18)	1.04 (0.94–1.15)
Very high	1.01 (0.89–1.15)	0.94 (0.83–1.07)
**Marginalization (CAN-Marg)**		
*Households and dwellings*		
Very low	Ref.	Ref.
Low	0.97 (0.88–1.06)	0.97 (0.88–1.07)
Moderate	0.96 (0.88–1.05)	0.99 (0.90–1.08)
High	0.98 (0.89–1.07)	1.00 (0.91–1.10)
Very high	**0.91 (0.82–1.00)**	0.94 (0.85–1.04)
*Material resources*		
Very low	Ref.	Ref.
Low	0.97 (0.90–1.05)	0.94 (0.87–1.03)
Moderate	1.05 (0.97–1.14)	1.02 (0.93–1.11)
High	1.01 (0.92–1.10)	0.98 (0.89–1.07)
Very high	0.96 (0.87–1.05)	0.96 (0.87–1.05)
*Age and labour force*		
Very low	Ref.	Ref.
Low	0.95 (0.89–1.01)	0.97 (0.90–1.03)
Moderate	**0.87 (0.81–0.94)**	**0.92 (0.85–0.99)**
High	**0.81 (0.74–0.88)**	**0.85 (0.78–0.93)**
Very high	**0.83 (0.73–0.94)**	**0.87 (0.76–0.99)**
*Immigration and visible minority*		
Very low	Ref.	Ref.
Low	1.07 (0.96–1.12)	1.05 (0.94–1.19)
Moderate	**1.15 (1.04–1.28)**	**1.16 (1.04–1.30)**
High	1.06 (0.95–1.17)	1.04 (0.93–1.16)
Very high	1.03 (0.92–1.15)	1.01 (0.90–1.13)
**Material Deprivation**		
Very low	Ref.	Ref.
Low	0.91 (0.85–0.97)	0.94 (0.87–1.01)
Moderate	**0.86 (0.80–0.93)**	**0.90 (0.83–0.97)**
High	**0.83 (0.76–0.90)**	**0.87 (0.80–0.95)**
Very high	**0.71 (0.64–0.79)**	**0.77 (0.70–0.86)**
**Social Deprivation**		
Very low	Ref.	Ref.
Low	0.97 (0.89–1.05)	0.98 (0.90–1.07)
Moderate	0.94 (0.86–1.01)	0.97 (0.89–1.06)
High	**0.92 (0.85–1.00)**	0.98 (0.90–1.07)
Very high	**0.84 (0.77–0.92)**	**0.90 (0.82–0.99)**
**Greenness**		
Very low	Ref.	Ref.
Low	0.98 (0.91–1.07)	0.97 (0.88–1.05)
Moderate	0.96 (0.88–1.03)	0.93 (0.86–1.01)
High	**1.13 (1.04–1.23)**	**1.10 (1.01–1.20)**
Very high	**1.14 (1.05–1.24)**	**1.11 (1.01–1.21)**

Model 1 adjusted for sex, age, ethnicity, and marital status. Model 2 adjusted for Model 1 + smoking status, alcohol consumption, BMI and physical activity. Bolded font indicate statistical significance *p* < 0.05.

**Table 3 nutrients-14-04882-t003:** Associations between meeting recommendations for fruit and vegetable consumption and neighbourhood environment stratified by urbanicity.

	OR (95% CI)	
	Rural	Urban
**Marginalization (CAN-Marg)**		
*Households and dwellings*		
Very low	Ref.	Ref.
Low	**0.75 (0.62–0.90)**	1.06 (0.95–1.18)
Moderate	**0.81 (0.68–0.98)**	1.05 (0.94–1.18)
High	**0.81 (0.67–0.97)**	1.08 (0.96–1.22)
Very high	0.80 (0.64–1.00)	0.99 (0.88–1.11)
**Material Deprivation**		
Very low	Ref.	Ref.
Low	1.09 (0.92–1.30)	**0.91 (0.83–0.98)**
Moderate	1.04 (0.88–1.24)	**0.86 (0.78–0.95)**
High	1.00 (0.83–1.20)	**0.85 (0.77–0.95)**
Very high	0.95 (0.78–1.16)	**0.72 (0.63–0.82)**

Estimates from fully adjusted model 2: sex, age, ethnicity, marital status, smoking status, alcohol consumption, BMI and physical activity. Bolded font indicate statistical significance *p* < 0.05.

## Data Availability

Data used in this analysis are available upon approval by the BC Generations Project.

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
