# Peer review of "Dietary Intake and the Neighbourhood Environment in the BC Generations Project"

_nutrients, 2022, doi:10.3390/nu14224882_

Round 1

Reviewer 1 Report

The authors have studied the influence of social factors like neighborhood environment  dietary intake a novel approach for assessment healthy food chaises.

The introduction reviews in details the present knowledge of the influence of socio-ecological factors and models in order to conceptualize health and influence on health determinants.

Materials and methods are well described as well as Results section. Small corrections are needed on Tables number –L 210 /Table 3/ L227 – /Table2/; same for L237, L244.

Discussion section corresponds well with the results. However the limitations of the detailed dietary data /340-342/ should be discussed in this section more thoroughly.

My recommendation is focused on the  Conclusions section – on the basis of the vast statistical data in the study stronger implications could be drawn about socio environmental factors that influence  consumption of fruit and vegetables as a model for health eating.

Author Response

The authors have studied the influence of social factors like neighborhood environment dietary intake a novel approach for assessment healthy food chaises.

The introduction reviews in details the present knowledge of the influence of socio-ecological factors and models in order to conceptualize health and influence on health determinants.

Thank you for your positive review and helpful feedback.

Materials and methods are well described as well as Results section. Small corrections are needed on Tables number –L 210 /Table 3/ L227 – /Table2/; same for L237, L244.

We have corrected all references to the Tables in the text.

Discussion section corresponds well with the results. However the limitations of the detailed dietary data /340-342/ should be discussed in this section more thoroughly.

We have revised the Discussion as suggested and now reads “Dietary intake was limited to fruit and vegetable consumption which although correlated with dietary quality (26), provides limited insight on overall dietary intake which may be more relevant for health and chronic disease prevention. Fruit and vegetable consumption was self-reported which may be prone to bias. Further, consumption was limited to the typical amount consumed at present which may not reflect earlier consumption patterns that are relevant for disease prevention.”

My recommendation is focused on the Conclusions section – on the basis of the vast statistical data in the study stronger implications could be drawn about socio environmental factors that influence consumption of fruit and vegetables as a model for health eating.

We have revised the manuscript’s conclusion based on the reviewers recommendation to strengthen the message: “The findings underscore the complexity of dietary intake, and the importance of considering potential influential factors of diet beyond the immediate food environment. When possible, characterizing socio environmental factors such as those identified herein; greenness, social and material deprivation when implementing interventions aimed at improving dietary intake may help facilitate success. The findings also suggest potential targets for policies and interventions to improve dietary intake as inequalities were particularly evident in neighbourhoods with lower access to and attainment of basic material needs.”

Reviewer 2 Report

This is a well presented report from a valid dataset.

Its findings are not particularly innovative, the interest is related to quantification of he associations in a specific BC population.

Numbers of rural subjects are relatively small, thus a test for heterogeneity between uraban and rural is indicated. If heterogeneity is not significant, Table 3 (reported as Table 4 in the text) can be deleted.

Author Response

Thank you for the positive comments. 

Please see lines 183-185 where we state that we tested for interactions between urban and rural areas "Individual-level income and education and urbanicity (urban areas with census metropolitan areas of at least 100,000 of which 50,000 or more live in the core versus urban areas or census agglomerations (36)) were explored as potential effect modifiers using cross-product terms in models."

And lines 233-235 where we state "Associations between fruit and vegetable consumption, the households and dwellings dimension of marginalization, and material deprivation were modified by urbanicity (pinteraction<0.05)." 

As a result, we present the findings in Table 3 stratified by urban/rural. We have also now corrected the reference to Table 3 (incorrectly referred to as Table 4 in the prior version).